# Essential Oils and Their Components Control Behaviour of Yellow Mealworm (*Tenebrio molitor)* Larvae

**DOI:** 10.3390/insects14070636

**Published:** 2023-07-14

**Authors:** Gabrielė Bumbulytė, Jurga Būdienė, Vincas Būda

**Affiliations:** Laboratory of Chemical and Behavioural Ecology, Nature Research Centre, Akademijos Str. 2, LT-08412 Vilnius, Lithuania; gabriele.bumbulyte@gamtc.lt (G.B.); jurga.budiene@gamtc.lt (J.B.)

**Keywords:** insect behaviour, repellent, stored-food insect, bioassay, pest control

## Abstract

**Simple Summary:**

This study investigated the use of essential oils (EOs) from different plants to control the behaviour of *Tenebrio molitor* L., a pest of grain and flour. Among the EOs tested, spearmint was the most effective repellent for mealworm larvae, followed by clove. East-Indian lemongrass, thymus, lavandin, and eucalyptus EOs showed lower repellent properties. Terpinene-4-ol and carvone, were the most effective repellents, while limonene, myrcene, and γ-terpinene had no significant activity.

**Abstract:**

Beetle *Tenebrio molitor* L. (Coleoptera, Tenebrionidae) is a well-known pest of grain and flour in food stores and grocery shops. Recently, commercial cultivation of the insect was started for human food and animal feed. Behaviour control of this insect using natural repellents is promising both for grain protection and commercial cultivation. We analysed if natural products of plant origin, namely essential oils (EOs), could be used for this purpose. Behavioural tests were performed using EOs of six plants: thymus (*Thymus vulgaris*), eucalyptus (*Eucalyptus globulus*), spearmint (*Mentha spicata*), lavandin (*Lavandula* × *hybrida*), East-Indian lemongrass (*Cymbopogon flexuosus*), and clove (*Eugenia caryophyllus*). The most effective repellent for mealworm larvae was EO of spearmint, moderate activity showed that of clove and the least repellent were EOs of lemongrass thymus and lavandin. EO of eucalyptus caused almost no or very low effect. Six of the most abundant compounds of the EOs were selected for testing. The most effective single compounds were terpinene-4-ol and carvone, low-effective cis-sabinene hydrates and those of no significant activity were limonene, myrcene and *γ*-terpinene.

## 1. Introduction

Grain products are among the main sources of human food and feed for domestic animals. During grain storage, 5–20% of the harvest is lost worldwide due to insects [1]. This is related to several insect species whose main source of feed is stored dry grains.

In warehouses, grain pests can reproduce throughout the year, causing damage and losses in both grains and their products, as well as deteriorating their quality. These pests can breed not only in storage facilities with stored products but also in empty warehouses, silos, grain transportation vehicles, and grain residues in cracks in the floors and walls. Pest-contaminated feed that enters farms is a source of further pest spread.

Chemical insecticides are used for grain protection [2]. However, using insecticides widely has caused many negative effects both on the environment and on human health. Therefore, search for alternative pest control methods is required, and the application of natural compounds, especially plant-based ones, could be of great importance.

The yellow mealworm (*Tenebrio molitor* L.) is among those economically important insect pests that damage grain and flour a lot. Therefore, this species was chosen for the search for environmentally friendly means suitable for behavioural control. It has been revealed that insects of this species are distinguished by relatively high biomass growth, its larvae are nutritious and a rich source of protein, and it can be easily and extensively cultivated for human food and animal feed. With the development and/or improvement of technologies for the cultivation of this insect species, data on behavioural control are relevant.

Understanding the biology, behaviour, and ecology of *T. molitor* as a pest is crucial for developing effective control measures to ensure grain protection and food safety. The aim of this paper was to search for essential oils (EOs) (harmless to the environment, safe for humans) and their components that affect the behaviour of *T. molitor* larvae.

## 2. Materials and Methods

### 2.1. Insects

*T. molitor* larvae used in the tests were obtained from a culture maintained at the Laboratory of Chemical Ecology and Behaviour, Nature Research Centre, since 2019. The larvae were reared on a diet consisting of a mixture of oat flakes (70%), wheat bran (20%), and dry yeast (10%), and provided with carrot slices. During rearing, a temperature of 25 °C, a relative humidity of 60%, and a 12:12 h light–dark cycle was maintained. For the experiments, medium-sized larvae (14–20 mm in length) were selected.

### 2.2. Essential Oils and Chemicals of and Identification of Individual Components

The essential oils (EOs) used in the tests were purchased from “JSC Naujoji Barmune”, a retailer of EOs in Lithuania. The following EOs were used: thyme (*Thymus vulgaris*), eucalyptus (*Eucalyptus globulus*), spearmint (*Mentha spicata*), lavandin (*Lavandula* × *hybrida*), East-Indian lemongrass (*Cymbopogon flexuosus*), and clove (*Eugenia caryophyllus*). All EOs were labelled with the common and scientific names of the plant species they were extracted from, as well as with series numbers and expiry dates. However, the origin of the plants used to extract the EOs was not indicated.

Analytical standards used in the experiment were purchased from different companies: (R)-(−)-carvone from Toronto Research Chemicals Inc. (Toronto, ON, Canada); limonene (sum of enantiomers, analytical standard, ≥98% (GC)), myrcene (analytical standard, ≥95% (GC)), and *γ*-terpinene (analytical standard, ≥97% (GC)) from Fluka (Charlotte, NC, USA); *cis*-sabinene hydrate (analytical standard, ≥97% (GC)) and terpinene-4-ol (sum of enantiomers, analytical standard, ≥95% (GC)) from Sigma-Aldrich (Darmstadt, Germany).

### 2.3. GC-MS Analysis and Compound Identification

The chemical composition of the EOs was determined using a Shimadzu GC/MS-Q2010 PLUS chromatograph (Kyoto, Japan), which was interfaced with a Shimadzu GC-MS-QP2010 ULTRA mass spectrometer (Kyoto, Japan). The system was equipped with a non-polar Rxi-5 Sil MS integra guard capillary column (30 m × 0.25 mm × 0.25 µm) from Restek, USA. The analyses were performed in splitless mode, and mass spectra were generated in electron impact mode at 70 eV with a mass range of 33–400 *m*/z and 0.97 scans per second. Initially, 1 µL of each EO was dissolved in 1 mL of hexane, and then 1 µL of the solution was injected into the GC-MS system.

The GC oven temperature was programmed to start at 50 °C for 1 min, then increased by 5 °C per minute until it reached 160 °C, held at 160 °C for 2 min, and then programmed to increase again to 250 °C at a rate of 10 °C/min. The final temperature was maintained for 4 min. Helium (He) was used as a carrier gas at a flow rate of 1 mL/min. The FID detector and injector temperatures were set at 250 °C, and the ion source temperature was set at 220 °C.

Compounds were identified based on a comparison of experimental retention time (RT) and retention indices (RI) with corresponding published data [3], as well as computer libraries of mass spectra, using “GC/MS solution” v. 2.71 software from Shimadzu and Wiley and NIST. The identification of a compound was approved if mass spectra library data matched computer data with a probability equal to or greater than 90%. The retention indices were calculated relative to the retention times of a series of n-alkanes (C8–C28) using the GC-MS program described above. The relative percentage composition of the EOs was computed based on GC peak areas without correction factors.

### 2.4. Behavioural Test

The behavioural assay was performed in a Petri dish using a zone-preferred method as illustrated in Figure 1. The dish was 10.5 cm in diameter and the bottom was covered with filter paper. To one side of the dish, a stimulus consisting of either an EO or a single chemical compound was applied 2–3 mm from the edge. On the opposite side of the dish, a solvent (control) was applied at the same distance from the edge as the stimulus. The EO or chemical compound was applied in five consecutive spots, each containing 2 µL, for a total of 10 µL. The solvent was applied in the same manner and volume. The concentration of the EO was 1 µL/mL in hexane, and a dose of 0.01 µL was used based on preliminary observations. For testing the effects caused by individual compounds, three doses of each were used: 0.01 mM, 0.1 mM, and 1 mM. Hexane was used as the solvent and served as a control. After presenting the stimulus for approximately 0.5 min to allow for solvent evaporation, a single mealworm larva was placed at the centre of the dish. The dish was then positioned under a dim light source, and an air exhaust was adjusted to prevent the accumulation of volatiles. The test was conducted at a temperature of 23 ± 2 °C.

The larva’s behaviour was recorded for five minutes using a video camera, and the footage was analysed with EthoVision XT 12, computer-based animal behaviour analysis software developed by Noldus in the Netherlands. Ten behavioural recordings were analysed for each stimulus tested (either EO or chemical compound), and each larva was used only once. The analysis of the mealworm larva’s behaviour in both the control and stimulus zones was based on two criteria: distance moved, and time spent in each zone.

### 2.5. Data Analysis

Statistical analysis of the results was conducted using Microsoft Excel (USA) and STATISTICA (USA) software programs. The distance moved and time spent in the alternative zones were analysed using the Wilcoxon matched-pairs test for both the effects of the EOs and the single compounds. A *p*-value less than 0.05 was considered statistically significant for the behavioural parameters in the alternative zones.

## 3. Results

### 3.1. Movement of Mealworm Larva in the Arena

Movement of mealworm larvae in the arena was recorded, and it was revealed that in the absence of any stimulation, they mostly stayed in the peripheral zone along the edges of the arena, as depicted in Figure 2. Therefore, their behaviour was further recorded only in the periphery of the arena, which was a 1.5 cm wide track referred to as a zone. Prior to testing, we assessed the distance moved by *T. molitor* larvae and the time spent in the left or right zones of the arena. The analysis showed no significant differences (Wilcoxon matched pairs test, *p* > 0.05).

### 3.2. Effect of Essential Oils

*(a) Time spent in zones:* While moving in the arena, *T. molitor* larvae spent significantly less time in the zones where either lemongrass, thyme, lavender, clove, or spearmint EO was presented, compared to the time spent in the alternative (control) zone, as shown in Figure 3. The recorded durations differed statistically significantly (*p* < 0.05), and depending on the EO studied, ranged from 4.87 to 15.47 times. However, the eucalyptus EO did not have a significant effect, as the time spent by the larvae in the control and stimulus zones did not differ significantly (*p* > 0.05), as presented in Figure 3. In total, the quantified behavioural data demonstrated that the effect of the studied EOs on the behaviour of *T. molitor* larvae varied from 1.84 (eucalyptus EO) to 5.74 times (peppermint EO).

*(b) Distance moved in zones:* The distance moved by the larvae in the zone with EO was significantly shorter (*p* < 0.05) compared to the control zone. The distance moved was 2.23 times shorter in the presence of eucalyptus EO, 5.15 times shorter in the presence of lavender EO, 5.29 times shorter in the presence of thyme EO, 6.47 times shorter in the presence of lemongrass EO, 12.76 times shorter in the presence of clove EO, and 17.24 times shorter in the presence of spearmint EO, compared to the control zone, as depicted in Figure 4. Notably, a qualitative change in larval behaviour was observed when exposed to lemongrass EO, as the larvae demonstrated escape behaviour: fell on their side, cringed, and curled.

When summarising the reactions of the mealworms, it should be noted that the effects varied depending on the type of EO used. The effects of each EO were evaluated based on the time spent (t) and distance moved (d) and were classified into three groups: weak effect (if the ratio of S/C changed from 0 to 6), moderate effect (if the ratio changed from >6 to 12), and strong effect (if the ratio changed from >12 to 18). The total effect caused by each EO was evaluated based on both criteria and presented in Table 1.

The behavioural analysis allows the classification of the EOs tested as having weak, moderate, or strong repelling effects on *T. molitor* larvae. Eucalyptus, lemongrass, thyme, and lavender EOs showed weak repelling effects, while clove EO exhibited moderate repelling effects and spearmint EO was strongly repelling.

To identify the chemical compounds in the EOs responsible for the observed behavioural reactions in *T. molitor* larvae, we conducted an analysis of the chemical composition of the EOs.

### 3.3. Chemical Composition of EOs

The number and abundance of volatile components identified in the EOs varied depending on the type of EO tested. Lavandin EO contained the highest number of compounds, with 76 detected, while clove and eucalyptus EOs had relatively fewer compounds, with 36 and 28, respectively. Only three compounds, limonene, linalool, and trans-caryophyllene, were common to all six EOs studied, with their abundance ranging from trace to almost 30%. In lavandin EO, linalyl acetate was the most abundant compound. Thyme EO contained *cis*-sabinene hydrate (36.55%) as the main compound, along with terpinen-4-ol (13.08%) and thymol (7.42%), which were absent in the other EOs analysed. Carvone (78.01%) was the major compound in spearmint EO, with a significant amount of limonene (11.31%). Lavandin EO contained linalyl acetate (39.78%), linalool (27.88%), and camphor (8.31%). More than 97% of clove EO was made up of eugenol (75.86%), eugenol acetate (11.64%), and *trans*-caryophyllene (9.84%). The major compound in eucalyptus EO was 1,8-cineole (86.56%), and only three other constituents were found in quantities greater than 1% of the total number of compounds identified: α-pinene (5.22%), *γ*-terpinene (3.97%), and *α*-terpineol (1.07%). East-Indian lemongrass EO was rich in geranial (42.07%) and neral (30.48%), with significant amounts of geraniol (6.31%) and geranyl acetate (4.08%). In all the EOs analysed, one or two compounds were predominant only, while others were present in considerably smaller or minor amounts (Table 2).

The analysis of the chemical composition of the EOs highlighted both predominant and minor compounds. For a more detailed study, compounds of spearmint (the most effective EO) were selected for behavioural testing: both predominant (carvone and limonene) and present in small quantities (myrcene, limonene, γ-terpinene, and cis-sabinene hydrate). Among them, two were predominant in thymus EO (cis-sabinene hydrate and γ-terpinene-4-ol), while the others co-occurred as components of up to 5 EOs analysed/tested (Table 2).

### 3.4. Effect of Single Compounds

To test the impact of individual chemical compounds, present in the EOs’ composition on the behaviour of *T. molitor* larvae, six compounds from spearmint EO (which exhibited high effectiveness) and thyme EO (which belonged to the group of EOs with low activity) were selected.

*Carvone.* This chemical compound has been found to have an impact on the behaviour of *T. molitor* larvae based on both tested parameters. The larvae moved significantly shorter distances (*p* < 0.05) in the stimulus zone than in the control zone, with a reduction factor of 1.76 at a dose of 0.1 mM and 5.02 at a dose of 1 mM. However, the lowest tested dose of 0.01 mM did not have any effect (see Figure 5A). In addition, the time spent by the larvae in the stimulus zone was significantly changed (*p* < 0.05) only at the highest tested dose of 1 mM, with a reduction factor of 3.62 (see Figure 5B).

*Limonene, myrcene, and γ-terpinene.* The distances moved and time spent by the larvae in both the control and stimulus zones were similar compared to the control and did not exhibit significant differences across all tested doses (0.01; 0.1; 1 mM) (*p* > 0.05).

*cis-Sabinene hydrate.* When evaluating the effect of this chemical compound, a statistically significant difference in activity within the stimulus and control zones was observed only for the distance moved at a dose of 1 mM (*p* < 0.05) (Figure 6A), with the distance in the control zone being 1.92 times longer than in the stimulus zone.

*Terpinene-4-ol.* The larvae’s behaviour was significantly impacted by all doses of 4-terpineol tested (0.01 mM, 0.1 mM, and 1 mM) (*p* < 0.05). The distances moved in the control zone were, respectively, 1.81, 3.60, and 2.06 times higher than in the stimulus zone (see Figure 7A).

Although the lowest dose of terpinene-4-ol tested (0.01 mM) did not cause any changes in the time spent in the zones, larvae avoided the zone and spent more time in the control zone by 4.21 and 3.61 times, respectively, when higher doses (0.1 mM and 1 mM) were presented (see Figure 7B). Terpinene-4-ol caused a qualitative change in the behaviour of the larvae, as they fell on their sides, cringed, and curled.

To summarise the larvae’s reactions to single chemical compounds tested based on two applied criteria (time spent (t) and distance moved (d) in C and S zones), the compounds were grouped according to their impact. The impact of each tested criterion was classified as low (if the C/S ratio changed from >0 to 2), average (if the ratio changed from >2 to 4), or high (if the ratio changed from >4 to 6). When evaluating the impact according to both criteria, the average value was taken into consideration (see Table 3).

Based on these results, it can be concluded that among the tested chemical compounds present in EOs, terpinene-4-ol has the most significant repellent effect on *T. molitor* larvae, followed by carvone.

## 4. Discussion

Until now, only the toxic effects of some of the tested EOs on the yellow mealworm, *T. molitor*, have been known, observed either in adults, larvae, or both stages of the beetle. There have been much fewer data on the impact on behaviour, especially at the larval stage.

For instance, topical application of spearmint EO on adult *T. molitor* beetles has been shown to cause toxic effects [4]. Similar effects have been observed in a few species of coleopterans, such as *Epicauta atomaria* (Germar), in both adults and larvae [5]. Our data on the behavioural effect of spearmint EO expand existing knowledge about the impact of this EO on *T. molitor* larvae.

Toxic effects of clove EO on *T. molitor* larvae and adults have been demonstrated by topical application of the EO in acetone on the thorax [6]. Our study expands this knowledge by demonstrating that clove EO is repellent for *T. molitor* larvae, resulting in reduced distance moved in the zone where the EO was presented on the substrate. Moreover, we observed qualitatively different behavioural reactions of the larvae compared to the effects of the other EOs tested, such as dropping, cringing, and curling. Such behaviour can be considered escape behaviour, as reported for larvae of *Drosophila melanogaster* [7]. Our findings on the effect of clove EO on *T. molitor* larvae are novel and provide an opportunity to identify compounds that induce escape behaviour.

The lavandin EO has been previously reported to have a weak repellent effect on *T. molitor* larvae, but this effect was observed after a long exposure period (3 h) only [8]. The method used in the present study allowed much faster detection of this reaction, within 5 min following application. The effect of *Cymbopogon flexuosus* EO on mealworms is unknown. EO of another plant species from the genus Cymbopogon, *C. citratus*, was not found to be repellent to mealworm larvae [9], but it was a strong repellent to another stored grain pest, *Tribolium castaneum* (adults) [10]. In this study, we report novel data on the repellent effect of *C. flexuosus* EO on *T. molitor* larvae, although it was weaker than the effect evoked by the EOs of the other plants studied.

The toxic effect of thyme EO on mealworm larvae and adult beetles has been reported following oil application to wheat feed, but it was not considered one of the strongest-acting oils [11]. The strong toxic effect of thyme EO was observed at very high doses (0.14 mg/cm^3^) only [4]. Thyme EO has also been found to have a repellent effect on other adult beetles, including *Sitophilus zeamais* [12], *Meligethes aeneus* [13], and *Ips typographus* [14]. However, the repellent effect of thyme EO on mealworm larvae has not been previously reported.

Previous studies have demonstrated that *Eucalyptus globulus* EO acts as a repellent for adult mealworms [15] but not for larvae [16]. However, dried and ground leaves of this eucalyptus were found to be repellent to larvae. Our results explain the contradiction reported in the study of Martynov et al., (2019): not only leaves but EO of *Eucalyptus globulus* as well is repellent for *T. molitor* larvae [16].

The composition of the EOs used for testing *T. molitor* larvae that caused behavioural reactions was evaluated, as the composition can vary depending on many factors. Thyme (*Thymus vulgaris* L.), a commonly used flavouring agent and medicinal herb, has several chemotypes, including those of linalool, borneol, geraniol, sabinene hydrate, thymol, carvacrol, and multiple other components’ chemotypes [17]. The most common chemotypes are thymol, carvacrol, geraniol, and linalool, along with multiple combinations of these compounds. However, borneol and sabinene hydrate chemotypes are considered rare [18,19]. The EO of *Thymus* used in our study belongs to the rare chemotype of sabinene hydrate.

Spearmint EOs are characterised by a high amount of carvone, which can account for over 50% of all identified compounds [20], accompanied by a significant amount of limonene, which can range from 9% to 22% [21]. Hussain et al. (2010) reported that the second most abundant compound in *M. spicata* EO from Pakistan was *cis*-carveol (24%), with limonene (5.3%) as the third most abundant compound [22]. The EO used in our study had a similar composition of predominant components to that reported by Chauhan et al. (2009) [21].

Lavandin EOs differ from true lavender EOs in terms of much higher amounts of non-major constituents, such as 1.8-cineole (4–7%) and camphor (6–11%). The lavandin EO used in this study, with linalool and linalyl acetate as major compounds, had a standard chemical profile [23].

It has been reported that eugenol, which can account for more than 50% of all identified constituents in clove EO, is usually accompanied by eugenol acetate, trans-caryophyllene, and *α*-humulene, which can comprise up to 90% of clove EO [24]. The clove EO used in this study contained the same predominant components but was slightly distinguished by the fact that they accounted for as much as 98.4% of all compounds.

The chemical profile of the *Eucalyptus globulus* EO used in this study, with a predominant compound 1.8-cineole and a significant amount of α-pinene, was typical and consistent with the publications [25].

The chemical composition of the lemongrass EO used for testing was similar to the typical composition, with high amounts of neral and geranial as major compounds. These two compounds accounted for over 60% of all identified constituents in the EO [26].

Concerning the data on behavioural effects caused by single chemical compounds from the EOs’ bouquet, including carvone, limonene, myrcene, *γ*-terpinene, terpinene-4-ol, and *cis*-sabinene hydrate, it should be noted that the data are very scarce for many insect species, and for the larvae of *T. molitor*, they were still absent prior to this study.

Carvone has been found to be toxic for some stored grain insect pests at various stages of development, ranging from eggs to adults [27,28]. Caraway (*Carum carvi* L.) EO, which contains approximately 68% carvone, has been shown to be toxic to *T. molitor* adults [29]. In the current study, we identified the repellent property of carvone, which was rated between average and high when compared to other repellent compounds (Table 3).

It is well known that terpinene-4-ol is repellent for adults of the lesser grain borer (*Rhyzopertha dominica*) [30]. This study reports for the first time on the repellent properties of this compound to *T. molitor* larvae, including its ability to evoke a special kind of behavioural reaction called escape behaviour. Interestingly, this compound is quite abundant in thyme EO (13.08%, Table 2), which did not cause behavioural reactions in the larvae (Table 1). Therefore, one can assume that this EO contains compounds that mask the repellent property of terpinene-4-ol.

The effect of *cis*-sabinene hydrate on *T. molitor* larvae behaviour was not known. However, there were reports on the toxic properties of EOs of *Origanum onites* and *O. vulgare* ssp. *hirtum*, which contain a high concentration of this compound (about 14.6%), to adults of the confused flour beetle (*Tribolium confusum*) and the lesser grain borer (*R. dominica*) [31]. As a potentially toxic but low-repellence compound, *cis*-sabinene may be promising for *Tenebrio* larvae control.

The effect of limonene on *T. molitor* larvae has not been studied to date. Moreover, there are no data on its toxicity, although lemon EO, which is rich in this compound is toxic to *T. molitor* larvae [9].

The data we have obtained indicate that *γ*-terpinene does not cause significant behavioural reactions in *T. molitor* larvae, which is consistent with the weak repellence of bergamot orange EO rich in this compound [8].

Similarly, the effect of myrcene on the behaviour of *T. molitor* larvae was not significant. However, myrcene is one of the major components of *Pistacia atlantica* ssp. *kurdica* EO, which has insecticidal activity against several species of *Tribolium* beetles when used as a fumigant [32]. Therefore, if myrcene were found to be toxic to *T. molitor* larvae (as well as other compounds with similar properties, i.e., limonene, *γ*-terpinene), it could be a promising agent for pest control purposes, as a toxic compound that does not cause avoidance reactions can be effective at low concentrations.

In summary, the results obtained regarding the behavioural effects of EOs from six plant species and six single chemical compounds present in their composition, which were tested on *T. molitor* larvae, could be useful for the search and development of non-insecticidal and environmentally friendly means for grain protection. Moreover, these results could be beneficial in the cultivation of *T. molitor* for feed or food, where the control of larval behaviour is needed.

## 5. Conclusions

Overall, these results highlight that the most effective repellent for mealworm larvae was the EO of spearmint and that of clove showed moderate activity. The EOs of lemongrass thymus, lavandin, and especially EO of eucalyptus were the least repellent. Six compounds of the spearmint EO were tested. The most effective single compounds were terpinene-4-ol and carvone, limonene, myrcene and *γ*-terpinene were low-effective *cis*-sabinene hydrates and had no significant activity.

## Figures and Tables

**Figure 1 insects-14-00636-f001:**
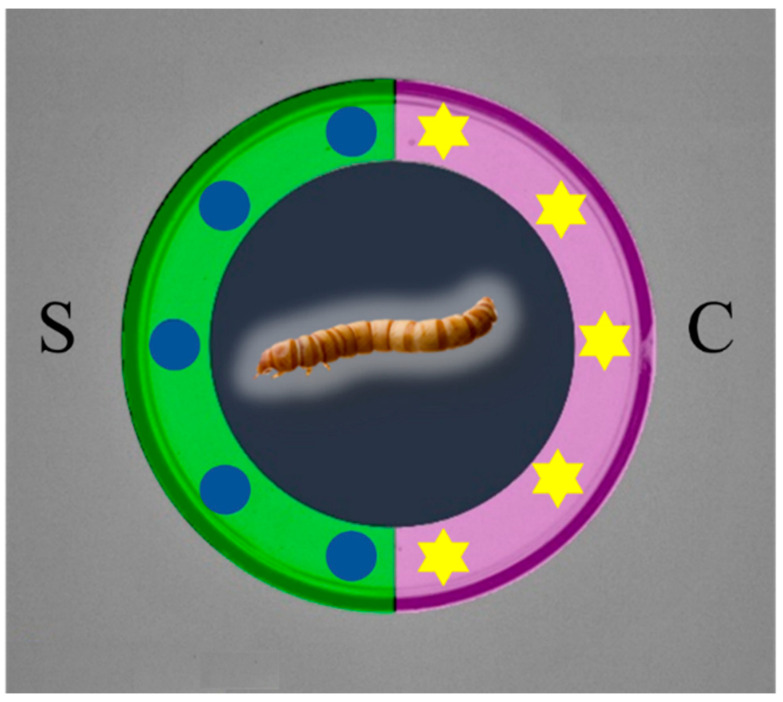
Schematic diagram of the two-choice assay in a Petri dish. The green strip indicates stimulus zone (S), the light violet strip indicates control zone (C); blue circles indicate spots where either EO or single chemical compound was applied; yellow stars indicate spots of solvent drops.

**Figure 2 insects-14-00636-f002:**
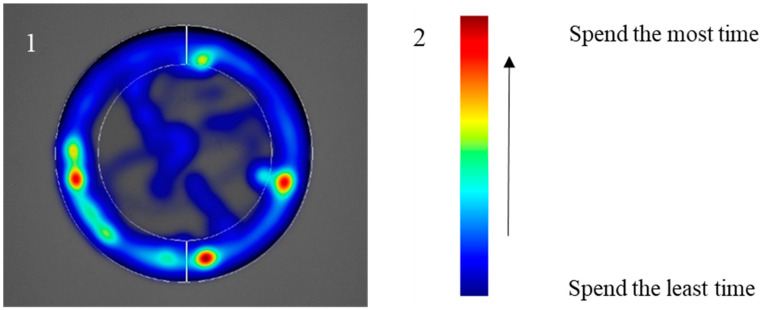
Heat map of *Tenebrio molitor* larvae movement in the arena. **1**—without stimulus (5 min, *n* = 10), **2**—Jet colour scale.

**Figure 3 insects-14-00636-f003:**
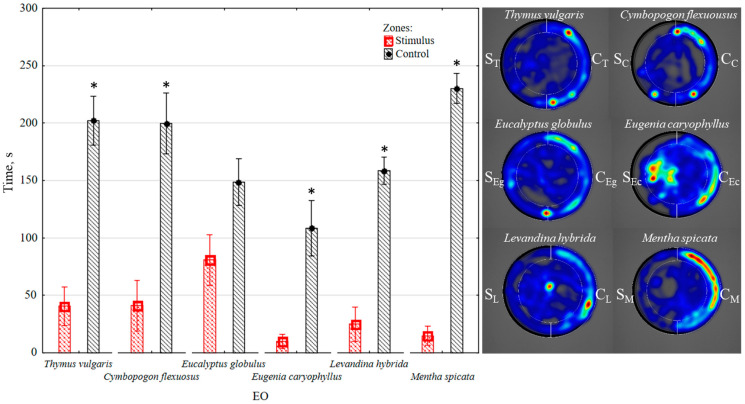
Average time spent by the mealworm (*Tenebrio molitor*) larva in alternative zones: control and containing thyme, lemongrass, eucalyptus, clove, lavender or spearmint EO (5 min, *n* = 10). Colour on heat maps as indicated in Figure 2. Statistically significant differences of duration in the stimulus and control zones are marked with an asterisk (*p* < 0.05) (Wilcoxon matched pairs test).

**Figure 4 insects-14-00636-f004:**
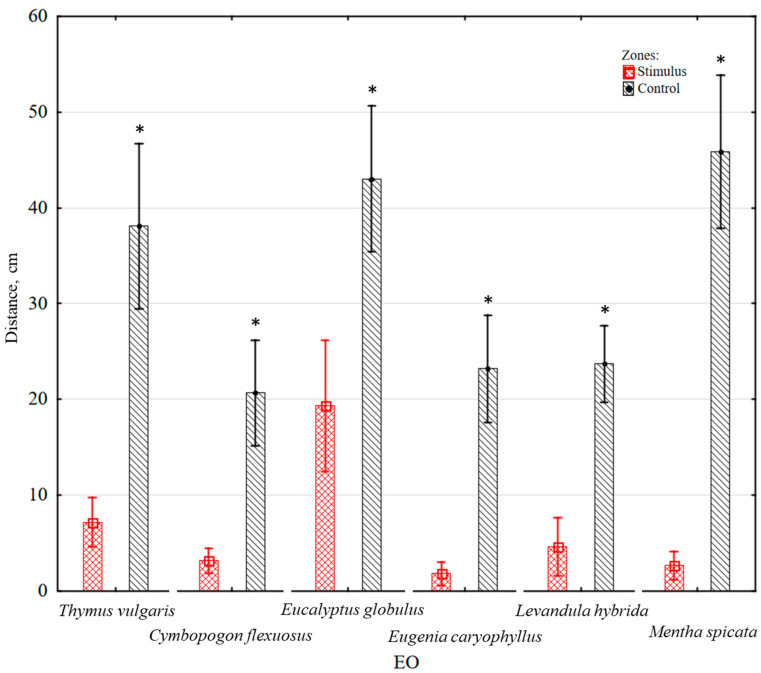
Average distance moved by mealworm (*Tenebrio molitor*) larvae in control and stimulus zone, containing either thyme, lemongrass, eucalyptus, clove, lavender or spearmint EO. Statistically significant differences in the stimulus and control zones are marked with an asterisk (*p* < 0.05) (Wilcoxon matched pairs test).

**Figure 5 insects-14-00636-f005:**
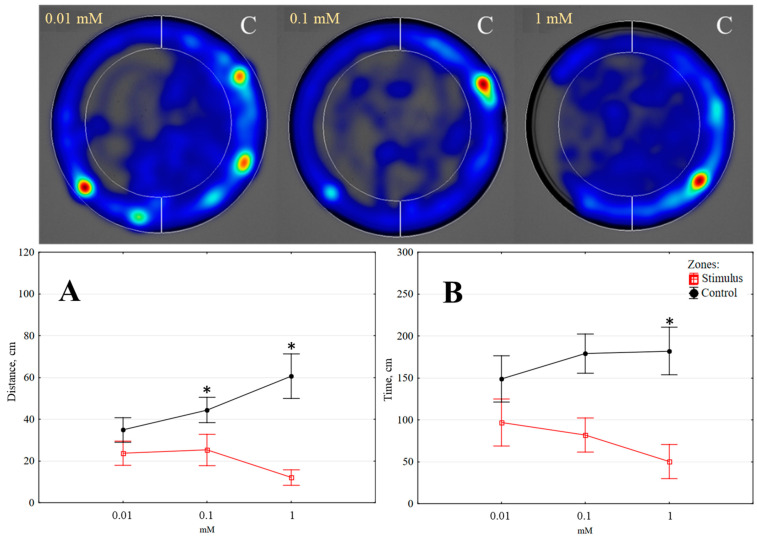
The distance moved (**A**) and time spent (**B**) by *Tenebrio molitor* larvae in alternative zones when exposed to different doses of carvone (5 min, *n* = 10); colour intensity on heat maps as indicated in Figure 2; C—control zone. Statistically significant differences in the stimulus and control zones are marked with an asterisk (*p* < 0.05) (Wilcoxon matched pairs test).

**Figure 6 insects-14-00636-f006:**
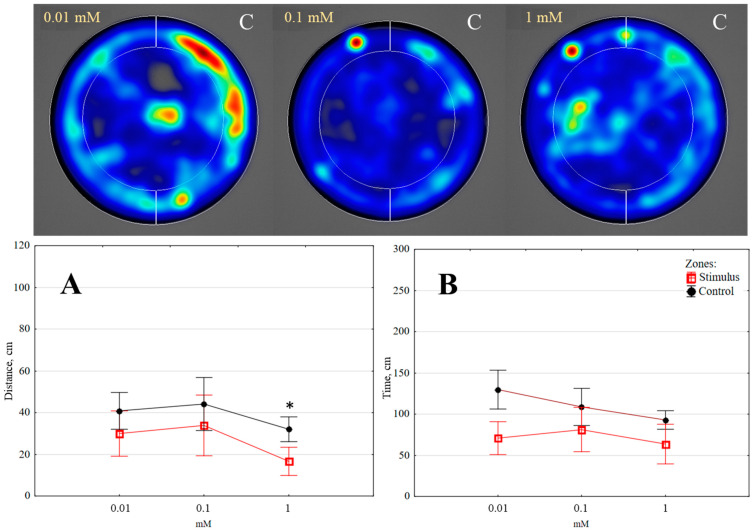
The distance moved (**A**) and time spent (**B**) by *Tenebrio molitor* larvae in the arena after exposure to different doses of *cis*-sabinene hydrate (5 min, *n* = 10); colour intensity on heat maps as indicated in Figure 2; C—control zone. Statistically significant differences in the stimulus and control zones are marked with an asterisk (*p* < 0.05) (Wilcoxon matched pairs test).

**Figure 7 insects-14-00636-f007:**
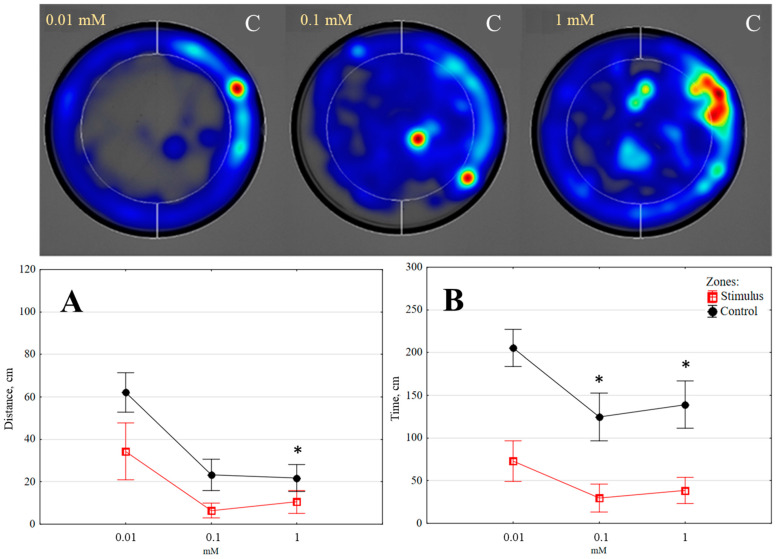
The distance moved (**A**) and time spent (**B**) by *Tenebrio molitor* larvae when exposed to different doses of 4-terpineol (5 min, *n* = 10); colour intensity on heat maps as indicated in Figure 2; C—control zone. Statistically significant differences in the stimulus and control zones are marked with an asterisk (*p* < 0.05) (Wilcoxon matched pairs test).

**Table 1 insects-14-00636-t001:** Effect of essential oil on *Tenebrio molitor* larvae behaviour based on the relative time spent and distance moved in stimulus and control zones.

	Distance Moved (d)		Time Spent (t)		(d + t)/2	
EO	S_d_/C_d_	Effect	S_t_/C_t_	Effect	(S_d_/C_d_ + St/C_t_)/2	Effect
*Cymbopogon flexuosus*	6.47 *	Average	4.87 *	Low	5.67	Low
*Thymus vulgaris*	5.29 *	Low	4.99 *	Low	5.14	Low
*Levandula hybrida*	5.15 *	Low	6.35 *	Average	5.75	Low
*Eugenia caryophyllus*	12.76 *	High	10.85 *	Average	11.80	Average
*Mentha spicata*	17.24 *	High	15.47 *	High	16.36	High
*Eucalyptus globulus*	2.23 *	Low	1.84	No	2.04	Low

* statistically significant differences (*p* < 0.05).

**Table 2 insects-14-00636-t002:** Main composition (% including constituents with quantity above 1.0% at least in one sample) of essential oils.

No	Compound Name	RI_Exp_/RI_Lit_	Thymus EO %	Spearmint EO %	Lavandin EO %	Clove EO %	Eucalyptus EO %	Lemongrass EO %
1	*α*-Pinene	939/939	0.95	0.26	0.23		**5.22**	0.12
2	Sabinene	975/975	**2.84**	0.11	0.06		0.09	
3	Myrcene	990/990	**3.75**	0.83	0.32		0.38	0.04
4	*α*-Terpinene	1017/1018	**2.89**				0.16	
5	*para*-Cymene	1024/1024	**6.00**	0.19	0.10		0.72	0.01
6	Limonene	1030/1029	**1.95**	**11.31**	0.50	0.16	tr.	0.24
7	1.8-Cineole	1031/1031	0.28	0.56	**4.15**		**86.56**	
8	*γ*-Terpinene	1059/1059	**6.52**	0.04	0.02		**3.97**	
9	*cis*-Sabinene hydrate	1071/1070	**36.55**	0.40				
10	Allyl hexanoate	1080/1079						**1.28**
11	Terpinolene	1088/1088	**1.02**	0.02	0.04		0.04	0.05
12	Linalool	1096	tr.	0.07	**27.88**	0.02	0.03	**1.18**
13	*trans*-Sabinene hydrate	1098/1099	**5.74**					
14	Camphor	1145/1146	0.15		**8.31**			
15	*cis*-Isocitral	1165/1164						**1.04**
16	Borneol	1169/1169	0.13		**2.39**			0.4
17	Terpinen-4-ol	1177/1177	**13.08**	0.63	**2.59**		0.25	
18	*trans*-Isocitral	1179/1180						**1.87**
19	*α*-Terpineol	1188/1188	**2.36**		**1.09**		**1.07**	0.19
20	*cis*-Dihydro carvone	1193/1192	0.14	**1.05**				
21	Neral	1239/1238					0.03	**30.48**
22	Carvone	1242/1243	0.07	**78.01**			0.03	
23	Linalyl acetate	1252/1257	0.26		**39.78**	0.01		
24	Geraniol	1257/1257			0.18			**6.31**
25	Geranial	1268/1268					0.04	**42.07**
26	Lavandulyl acetate	1289/1290			**3.28**			
27	Thymol	1290/1290	**7.42**					
28	Eugenol	1359/1359				**75.86**		0.04
29	Geranyl acetate	1380/1379			0.58			**4.08**
30	*β*-Bourbonene	1388/1388	0.03	**1.56**	0.03	0.01		
31	*trans*-Caryophyllene	1420/1419	**2.26**	0.80	**1.77**	**9.84**	tr.	**1.51**
33	*α*-Humulene	1454/1454	0.07			**1.09**		0.16
34	*γ*-Cadinene	1512/1513	0.01		0.23			**1.52**
35	Eugenol acetate	1520/1521				**11.64**		

RI_Exp_: Kovat’s indices determined experimentally on the non-polar column Rxi-5 Sil MS integra guard; RI_Lit_: Kovat’s indices for non-polar column Rxi-5 Sil MS integra guard from the literature [3]; tr—trace amount of the compound; relative percentage of compounds above 1% are marked in bold.

**Table 3 insects-14-00636-t003:** Effect of single compounds on *Tenebrio molitor* larvae behaviour based on the relative time spent and distance moved in stimulus and control zones.

		Distance Moved (d)		Time Spent (t)		(d + t)/2	
Chemical Compound	Dose (mM)	C_d_/S_d_	Effect	C_t_/S_t_	Effect	((C_d_/S_d_) + (C_t_/S_t_))/2	Effect
Carvone	0.01	1.47	No	1.54	No	1.51	Low
	0.1	1.76 *	Low	2.18	No	1.97	Low
	1	5.02 *	High	3.62 *	Average	4.32	High
Limonene	0.01	0.85	No	0.77	No	0.81	No
	0.1	1.17	No	1.29	No	1.23	No
	1	1.24	No	1.55	No	1.40	No
Myrcene	0.01	0.98	No	1.21	No	1.10	No
	0.1	1.09	No	1.31	No	1.20	No
	1	0.97	No	0.92	No	0.95	No
*γ*-Terpinene	0.01	1.13	No	1.18	No	1.155	No
	0.1	1.25	No	1.37	No	1.31	No
	1	0.93	No	1.04	No	0.985	No
*cis*-Sabinene hydrate	0.01	1.37	No	1.83	No	1.6	No
	0.1	1.30	No	1.34	No	1.32	No
	1	1.92 *	Low	1.46	No	1.69	Low
4-Terpineol	0.01	1.81 *	Low	2.82	No	2.32	Low
	0.1	3.60 *	Average	4.21 *	High	3.91	Average
	1	2.06 *	Average	3.61 *	Average	2.84	Average

* statistically significant difference (*p* < 0.05).

## Data Availability

The data presented in this study are available in the article.

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
