# Peer review of "Essential Oils and Their Components Control Behaviour of Yellow Mealworm (Tenebrio molitor) Larvae"

_insects, 2023, doi:10.3390/insects14070636_

Round 1
Reviewer 1 Report
An interesting paper with a unique perspective on using essential oils. The manuscript, including references and citations, must be formatted in the correct journal style. No need to spell out the genus name in every paragraph.
Authors should consider using a known repellent as a control. I would suggest either DEET or pyrethrins. The authors seem to assume that the most common essential oils with the highest percentage composition are the most repellent. It is certainly possible the minor components are the most repellent. Has this been examined?
There as many different insect species that have been tested using essential oils. The paper would be much better if some comparisons were made between your results and those with different beetle species and against other insect groups such as ants, bed bugs, and cockroaches. Are there any similarities or differences?
English needs to be reviewed by a native speaker. There are many awkward sentences and examples of incorrect usage.
Author Response
Response to Reviewer 1 Comments
Point 1: An interesting paper with a unique perspective on using essential oils. The manuscript, including references and citations, must be formatted in the correct journal style.
Response 1: The manuscript, including references and citations, was formatted in the journal style.
Point 2: No need to spell out the genus name in every paragraph.
Response 2: Corrected on lines 268, 271, 276, 285.
Point 3: Authors should consider using a known repellent as a control. I would suggest either DEET or pyrethrins.
Response 3: The suggestion do not require correction of the text. Extra research may be the goal of another paper.
Point 4: The authors seem to assume that the most common essential oils with the highest percentage composition are the most repellent. It is certainly possible the minor components are the most repellent. Has this been examined?
Response 4: Among the large number of minor components, there definitely can be active ones; it is a study requiring a separate analysis. The remark did not require expanded discussion; no correction of the text was made.
Point 5: There as many different insect species that have been tested using essential oils. The paper would be much better if some comparisons were made between your results and those with different beetle species and against other insect groups such as ants, bed bugs, and cockroaches. Are there any similarities or differences?
Response 5: We had no intention of discussing the results we had obtained from the proposed point of view. No corrections were suggested.
Reviewer 2 Report
The manuscript titled “Essential oils and their components control behaviour of yel- 2 low mealworm (Tenebrio molitor) larvae” reports the chemical composition and the bioactivity of six different essential oils and some compounds against Tenebrio molitor larvae. The manuscript introduces new and interesting techniques, however, it suffers some methodological problems and it cannot be accepted for publication in this form. It needs major revisions and also requires a revision of the English language.
In the detail the major concerns are:
1) lines 165-169. How did the Authors proceed to create the scale of high, medium and low activity ie, what is this scale based on? why 0-6 and not 0-5 or 0-7 to indicate a low bioactivity? Moreover, this approach excludes that the compounds can be attractive but the EOs it is now known that they can be repellent but in certain doses also attractive.
2) Lines 207-209: I cannot understand the logic of the choice of the different compounds tested which must be explained. In case the Authors have not a valid reason, they must complete the tests with the main compounds of each EO not yet tested, i.e.:
1.8 cineole for the bay
Eugenol for the clove
Geranial for lemongrass
Linalyl acetate for lavender
Cis sabinene hydrate for thyme and carvone for mint EOs are already present.
3) Why did the Author use only one dose (0,01 µL) for the EOs and 3 different doses for the compounds? Moreover, they do not mentioned any preliminary tests to identify the suitable dose choised .
4) Figure 3 and Figure 4. Please specify the meaning of the bars in the caption. Moreover which is the meaning of “Marking with an asterisk as Figure 3”??
Figure 5. in the caption, which is the meaning of “markings with an asterisk as Figure 3. Marking with an asterisk as Figure 2.”?? Please specify.
5) The manuscript contains some inaccuracies and needs to be checked and revised. Only as examples:
Figure 4. In the caption please correct Lavandina with Lavandula
Line 95. It is probably more correct to call “area preference method” the two-choice assay
Line 141. Which differences? Please specify.
Line 186. Linalyl acetate, not linalool
Line 35…. analysedknow
Lines 197-198 All essential oils have some major compounds and other minor compounds. The sentence is to be deleted.
6) The English needs in my opinion a revision
Author Response
Response to Reviewer 2 Comments
Point 1: The manuscript titled “Essential oils and their components control behaviour of yel- 2 low mealworm (Tenebrio molitor) larvae” reports the chemical composition and the bioactivity of six different essential oils and some compounds against Tenebrio molitor larvae. The manuscript introduces new and interesting techniques, however, it suffers some methodological problems and it cannot be accepted for publication in this form. It needs major revisions and also requires a revision of the English language.
In the detail the major concerns are:
Lines 165-169. How did the Authors proceed to create the scale of high, medium and low activity ie, what is this scale based on? why 0-6 and not 0-5 or 0-7 to indicate a low bioactivity? Moreover, this approach excludes that the compounds can be attractive but the EOs it is now known that they can be repellent but in certain doses also attractive.
Response 1: The scale was based on the ratio of activity in stimulus (S) and control (C) zones. As S/C ratio varied to 17.24 (Table 1), the interval was approximated and divided into 3 equal parts, thus steps from 0 to 6, and from ˃6 to 12, and from ˃12 to 18 were obtained. Attractivity of any EO tested was not recorded, although the method and apparatus we used were suitable for this.
This remark did not require correction of the text, it was not made.
Point 2: Lines 207-209: I cannot understand the logic of the choice of the different compounds tested which must be explained. In case the Authors have not a valid reason, they must complete the tests with the main compounds of each EO not yet tested, i.e.:
1.8 cineole for the bay
Eugenol for the clove
Geranial for lemongrass
Linalyl acetate for lavender
Cis sabinene hydrate for thyme and carvone for mint EOs are already present.
Response 2: For testing, we selected those rich components from EOs, which were available and were of high purity.
The remark did not require correction of the text, and it was not made.
Point 3: Why did the Author use only one dose (0,01 µL) for the EOs and 3 different doses for the compounds? Moreover, they do not mention any preliminary tests to identify the suitable dose choised.
Response 3: The dose of 0,01 µL for the EOs was chosen based on preliminary observations. One-dose- application provided sufficient results for comparative analysis of the effect caused by different EOs.
Text “based on preliminary observations” was added on line 107.
Point 4: Figure 3 and Figure 4. Please specify the meaning of the bars in the caption. Moreover, which is the meaning of “Marking with an asterisk as Figure 3”??
Response 4: The meaning of the bars was specified, Figure 3 corrected “Time, s” and Figure 4 corrected “Distance, cm” on the y-axes. “Marking with an asterisk as Figure 3” corrected as “Statistically significant differences of duration in the stimulus and control zones are marked with an asterisk (p<0,05) (Wilcoxon matched pairs test).”
Point 5: Figure 5. in the caption, which is the meaning of “markings with an asterisk as Figure 3. Marking with an asterisk as Figure 2.”?? Please specify.
Response 5: “markings with an asterisk as Figure 3. Marking with an asterisk as Figure 2.” corrected as “Statistically significant differences of duration in the stimulus and control zones are marked with an asterisk (p<0,05) (Wilcoxon matched pairs test).” on Figure 5.
Point 6: The manuscript contains some inaccuracies and needs to be checked and revised. Only as examples:
Figure 4. In the caption please correct Lavandina with Lavandula
Response 6: Caption corrected on Figure 4.
Point 7: Line 95. It is probably more correct to call “area preference method” the two-choice assay
Response 7: Corrected as “zone preference method”, see line 100.
Point 8: Line 141. Which differences? Please specify.
Response 8: Corrected as follows: The recorded durations differed statistically significant. Line 147.
Point 9: Line 186. Linalyl acetate, not linalool
Response 9: Corrected to “linalyl acetate”, line 193.
Point 10: Line 350…. Analysedknow
Response 10: corrected as “know” on line 353.
Point 11: Lines 197-198 All essential oils have some major compounds and other minor compounds. The sentence is to be deleted.
Response 11: The sentence indicates that one or two compounds dominated (in theory, there could be more). Therefore, we added word “only” on line 205, and believe that it is appropriate to leave slightly modified sentence.
Point 12: The English needs in my opinion a revision
Response 12: A native English editor revised the text, and his suggestions were added on lines 14, 20, 21, 66, 67, 315, 333, 336, 339, 348, 356.
Round 2
Reviewer 1 Report
A little additional polishing of the English would be helpful.
A little additional polishing would be helpful.
Author Response
Response to Reviewer 1 Comments
Point 1: A little additional polishing of the English would be helpful.
Response 1: Text revised and English checked.
Thank you for your feedback and for noting the need for additional polishing of the English in the manuscript. I greatly appreciate your suggestions and will make the necessary improvements to enhance the clarity and accuracy of the language throughout the manuscript. Your input is valuable, and I am thankful for your thorough review.
Reviewer 2 Report
Lines 207-209: understand that the Authors have used the compounds available in their lab but it is easy to refute their claims (compounds unavailable?) by even consulting the Sigma catalogue. It would be better to indicate a valuable reason for the choice of the compounds.
Author Response
Response to Reviewer 1 Comments
Point 1: Lines 207-209: understand that the Authors have used the compounds available in their lab but it is easy to refute their claims (compounds unavailable?) by even consulting the Sigma catalogue. It would be better to indicate a valuable reason for the choice of the compounds.
Response 1: Text “For a more detailed study compounds of spearmint (the most effective EO) were selected for behavioural testing: both predominant (carvone and limonene) and present in small quantities (myrcene, limonene, γ-terpinene and cis-sabinene hydrate). Among them, two were predominant in thymus EO (cis-sabinene hydrate and γ-terpinene-4-ol), while the others co-occurred as components of up to 5 EOs analyszed/tested (Table 2).” was added on line 219.
I would like to express my sincere gratitude for your careful evaluation and insightful comments on the notes provided for my manuscript. Your expertise and attention to detail have been immensely valuable in improving the overall quality. Your thoughtful suggestions and constructive feedback have helped me refine and strengthen various aspects of the manuscript. I genuinely appreciate the time and effort you have dedicated to reviewing and providing such valuable input.